# FedSLS: Exploring Federated Aggregation in Saliency Latent Space

## Abstract

Federated Learning (FL) is an emerging direction in distributed machine learning that enables jointly training a global model without sharing the data with server. However, data heterogeneity biases the parameter aggregation at the server, leading to slower convergence and poorer accuracy of the global model. To cope with this, most of the existing works involve enforcing regularization in local optimization or improving the model aggregation scheme at the server. Though effective, they lack a deep understanding of cross-client features. In this paper, we propose a saliency latent space feature aggregation method (FedSLS) across federated clients. By Guided BackPropagation (GBP), we transform deep models into powerful and flexible visual fidelity encoders, applicable to general state inputs across different image domains, and achieve powerful aggregation in the form of saliency latent features. Notably, since GBP is label-insensitive, it is sufficient to capture saliency features only once on each client. Experimental results demonstrate that FedSLS leads to significant improvements over the state-of-the-arts in terms of accuracies, especially in highly heterogeneous settings. For example, on CIFAR-10 dataset, FedSLS achieves 63.43% accuracy within the strongly heterogeneous environment $\alpha = 0.05$, which is 6% to 23% higher than the other baselines.

## CCS Concepts

• **Computing methodologies** → **Artificial intelligence**; *Distributed artificial intelligence*; Cooperation and coordination.

## Keywords

Latent Space, Federated Learning, Guided BackPropagation

## 1 Introduction

Federated Learning (FL) as a distributed machine learning paradigm, has garnered widespread exploration within computer vision [9, 21, 22, 29, 44], medical analysis [6, 17], and data security [39]. However, an inherent challenge facing FL is data heterogeneity. Specifically, different clients collect data based on diverse preferences, exhibiting independently and identically distribution and imbalance. While each participant optimizing towards a local empirical risk minimum, which is inconsistent with the global model. Accordingly, the averaged global model inevitably suffers from slow convergence speed and poor performance. Popular efforts

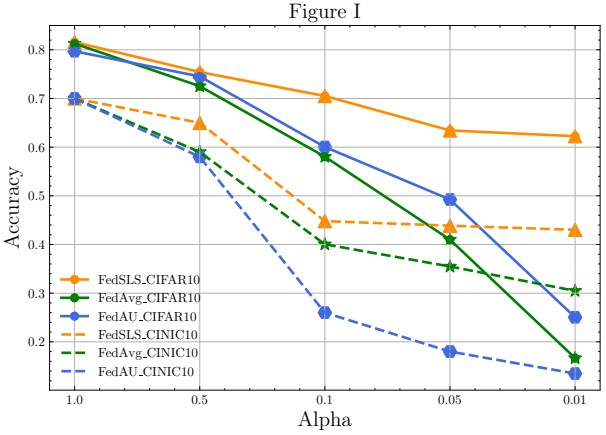

Figure I

**Figure 1: Improvement of our proposed approach FedSLS over FedAvg [30] and FedAU [42] under going from the less heterogeneous distribution to the strongly heterogeneous distribution. All methods are experimented on CIFAR-10 and CINIC-10, respectively.**

dedicated to addressing the data heterogeneity include data sharing [7, 8, 14, 49], client drift mitigation [1, 18, 22, 24] and aggregation scheme [12, 26, 40, 41, 46].

Among them, aggregation scheme is a fruitful avenue of explorations. Typical works either design novel strategies to enhance the aggregation phase [41, 48], or adaptively adjust aggregation using learned local information as a prior [3, 42, 47]. Then some researches can apply other rules to update the global model, such as momentum rules and retraining classifier by virtual feature [12, 28, 35, 38]. However, new aggregation strategies usually bring new pitfalls, such as FedKNN needs to record historical gradients, which increases the risk of privacy leakage and memory burden. In the meantime, they do not take fairness into account, that leading to poor generalization performance of the global model on some clients. In contrast, efforts adaptively adjusting aggregation usually take fairness into account so minimizing client discrimination. In this paper, we focus on adaptive aggregation adjustment to dig deeper into the latent contribution of each client to the aggregation stage, aiming to effectively cope with data heterogeneity.

We propose a saliency feature aggregation method FedSLS across multi-client, exploring optimal aggregation weights in saliency latent space. Leveraging Guided BackPropagation (GBP) [37], we transform deep models into powerful and flexible visual fidelity encoders, providing a lower-dimensional representational space which is perceptually equivalent to the data space called saliency latent space. In this space, feature achieves a near-optimal balance between reducing complexity and preserving detail, resulting in greatly improved saliency. As a result, FedSLS unfolds a consistent characterization of the data across multiple clients and uses saliency

**Figure 2: Architecture illustration of Guided Encoder$(\cdot)$ with an image $i \in D_k$ on client $k$. Before saliency embedding, we pre-train or fine-tune initialed model $\theta_k^0$ with local data $D_k$ first, making the parameters adapt to the task. Subsequently, saliency embedding yields all shallow-to-deep saliency latent variables $\sum G_{i,l,k}$ through trained model $\theta_k'$. Finally, we assign larger weights to the shallow saliency latent variables and smaller weights to the deeper saliency latent variables to obtain the saliency weights $R_k$ on client $k$ by means of the feature encoder. See more in Algorithm 1.**

features with high visual fidelity for high-performance aggregation optimization. Our contributions can be summarized as follows:

- We pioneeringly transform deep models into high visual fidelity encoders with GBP. It finds the optimal balance between reducing complexity and preserving detail in the low-dimensional latent space, which greatly improves feature saliency of the latent variables.
- We propose FedSLS, a saliency federated learning method to enable high-performance aggregation leveraging saliency latent space feature. To the best of our knowledge, our algorithm is the first work to explore cross-client consistency representations in latent space.
- Our extensive experiments demonstrate that FedSLS not only outperforms existing state-of-the-art federated learning methods but also shows remarkable performance in highly heterogeneous environments.
- We conduct multiple evaluations of the original images, saliency latent variables, and vanilla latent variables[1] across different similarity metrics, demonstrating that our method broadly enhances visual fidelity.

## 2 Related Work

Federated learning [16, 23, 30] is a fast-growing research field and remains many open problems to solve. In this work, we focus on addressing the non-IID quagmire [11, 49]. Relevant works have pursued the following three directions.

### 2.1 Client Drift Mitigation

FedAvg [30] has been the *de facto* optimization method in the federated setting. However, when it is applied to the heterogeneous setting, one key issue arises: when the global model is optimized with different local objectives with local optimums far away from each other, the average of the resultant client updates (the server update) would move away from the true global optimum [18]. The cause of this inconsistency is called 'client drift'. To alleviate it,

---

[1]The vanilla latent variables are obtained by conventional works, such as Variational AutoEncoders (VAE) [19].

FedAvg is compelled to use a small learning rate which may damage convergence, or reduce the number of local iterations which induces significant communication cost [25]. There have been a number of works trying to mitigate 'client drift' of FedAvg from various perspectives. FedProx [24] proposes to add a proximal term to the local objective which regularizes the euclidean distance between the local model and the global model. MOON [22] adopts the contrastive loss to maximize the agreement of the representation learned by the local model and that by the global model. SCAFFOLD [25] performs 'client-variance reduction' and corrects the drift in the local updates by introducing control variates. FedDyn [15] dynamically changes the local objectives at each communication round to ensure that the local optimum is asymptotically consistent with the stationary points of the global objective. FedIR [13] applies importance weight to the local objective, which alleviates the imbalance caused by non-identical class distributions among clients.

### 2.2 Data Sharing

The key motivation behind data sharing is that a client cannot acquire samples from other clients during local training, thus the learned local model under-represents certain patterns or samples from the absent classes. The common practices are to share a public dataset [49], synthesized data [8, 14] or a condensed version of the training samples [7] to supplement training on the clients or on the server. This line of works may violate the privacy rule of federated learning since they all consider sharing raw input data of the model, either real data or artificial data.

### 2.3 Aggregation Scheme

A fruitful avenue of explorations involves improvements at the model aggregation stage. These works are motivated by obtaining a federated model that generalizes well across clients. Many of them differ from the weighted aggregation parameters to obtain a global model. FedAvgM [12] applies the momentum rule to update the global model, which can improve robustness to heterogeneous distributed data, and FedNova [41] eliminates inconsistencies by normalizing local updates before averaging them. Subsequently,

some researches can apply other rules to update the global model. CCVR [28] and CReFF [35] illustrate that the heterogeneity of the classifier is the main reason for the performance degradation of models trained on non-IID data. In addition, some studies have suggested that adaptive aggregation adjustment can also improve the performance of the global model. FedAU [42] estimates the optimal aggregation weights based on historical aggregation information while taking into account the level of clients participation.

## 3 Method

### 3.1 Preliminaries

In federated learning, an optimization problem we need to solve is:

$$\arg \min_{\theta \in \mathbb{R}^n} [\ell(\theta) \triangleq \frac{1}{m} \sum_{k=1}^m L_k(\theta)], \qquad (1)$$

where $L_k(\theta) \triangleq \mathbb{E}_{(x,y) \sim P_k} [\ell(\theta, (x, y))]$ is the empirical loss of client $k$, $P_k$ is the data distribution for client $k$ across $K$ clients.

A popular approach to solve Eq. 1 in federated settings is FedAvg [30]. At each round, a subset of clients is selected (typically randomly) and the server broadcasts its model to each client. In parallel, the clients run SGD on their own loss function $\ell^k$ and then send their updated models to the server.

---

**Algorithm 1** Guided Encoder in FedSLS

---

1: **Input:** client $k$; training dataset $D_k$; initial parameter $\theta_k$.
2: **Output:** saliency weight $R_k$.
3: **for** each client $k \in K$ **do**
4:      **for** image $i \in D_k$ **do**
5:          *# forward propagation*
6:          $Y = P(\theta; x)$.
7:          *# back propagation*
8:          **for** convolutional layer $l = 1$ **to** $L$ **do**
9:              *# saliency embedding*
10:             $G_{i,l,k} = \frac{\partial F_{i,l,k}}{\partial Y} \cdot \max(0, F_{i,l,k})$.
11:             *# average in channel dimension*
12:             $\bar{G}_{i,l,k} = \frac{1}{C} \sum_{c=1}^C G_{i,l,k}^{(c)}$.
13:             *# L2 paradigm computation*
14:             $N_{i,l,k} = \sqrt{\sum \|\bar{G}_{i,l,k}\|_2}$.
15:         **end for**
16:         $R_{i,k} = \sum_{l=1}^N w_l \cdot N_{i,l,k}$.
17:     **end for**
18:     *# client $k$ saliency weight computation*
19:     $R_k = \sum_{i \in D_k} R_{i,k}$.
20: **end for**

---

The server then updates its model to be the average of these client models. Suppose that at the $r$-th round, the server has model $\theta$ and samples a client set $S_k$. Here we use a standard gradient descent form to update parameters:

$$\theta \leftarrow \theta - \eta \Delta, \qquad (2)$$

where $\Delta$ is the aggregated clients' gradients and $\eta$ is the learning rate of the server, which is typically 1.0. FedAvg uses weighted

average aggregation to compute $\Delta$. Then we can write weighted average aggregation as:

$$\Delta = \sum_{k \in S_k} (w_k \cdot \Delta_k), \qquad (3)$$

where $\Delta_k = \theta - \theta_k$ is the accumulated gradients within a training round of client $k$, and $w_k = |D_k|/\sum_{k \in S_k} |D_k|$ is the aggregated weight of client $k$ across the activated clients $S_k$. $D_k, \theta_k$ are the training dataset and trained parameters on client $k$, respectively.

### 3.2 Departure to Saliency Latent Space

Our core perspective lies in transforming deep models to high visual fidelity encoders which embedding the data into saliency latent space. We define an encoder $\hat{\mathcal{E}}(\cdot)$ that embeds all images on the client into the latent space and quantifies the value of these latent variables containing saliency feature information as $R_k$. In this paper, we have redefined the aggregated weight $w_k$ as $R_k/\sum_{k \in S_k} R_k$. Thus, our main challenge is to design an appropriate coder $\hat{\mathcal{E}}(\cdot)$ that can maximize the features extracted from the data.

Conventional unsupervised feature embedding methods seek to encapsulate the semantic or structural information of data. Formally, given an image $x \in \mathbb{R}^{H \times W \times 3}$, the encoder $\mathcal{E}$ transmutes $x$ into a latent vector $z = \mathcal{E}(x)$ by compressing the spatial dimensions by a factor $f = H/h = W/w$, thus $z \in \mathbb{R}^{h \times w \times c}$, where typically $f = 2^m$ for some $m \in \mathbb{N}$. As a rule of thumb, $m$ is 1, 2 and 3. However, such methods are criticized for lossy compression while they manage to preserve significant semantic content in the compressed latent space $z \in \mathbb{R}^{h \times w \times c}$, the original spatial details of images are compromised. They incline towards an overarching semantic representation at the expense of local nuances, thereby not serving as an optimal method for feature extraction. Furthermore, the encoder $\mathcal{E}(\cdot)$ fails to discern and discard irrelevant and noisy features which are not conducive to learning, resulting in both essential and superfluous information being coalesced into the latent representation $z = \mathcal{E}(x)$, thus hindering the processing of features in lower-dimensional spaces.

Drawing inspiration from [37] on striving for simplicity, we propose the utilization of GBP to transform deep models into encoders capable of preserving high visual fidelity. Saliency feature maps generated during the GBP can granularity concentrate on local object information. Moreover, saliency latent variables, formulated within a supervised paradigm, can reflect the model's inner features, significantly curtailing the emphasis on non-essential and redundant details.

In order to verify our idea, we designed a series of experiments. The first one is to directly compare the amount of feature information exhibited by saliency latent variables and vanilla latent variables by calculating the similarity with the original images. We then utilize vanilla latent variables instead of saliency latent variables in the FedSLS framework to guide federated aggregation. We call this algorithm FedSL and compare its experimental results with FedSLS to illustrate the validity of our argument. Further elaboration is provided in Section 4. Our empirical findings affirm the superior semantic and structural content of saliency latent variables, an approach we term **Saliency Embedding**, with the resulting embedding domain referred to as the **Saliency Latent Space**.

### 3.3 FL with Saliency Latent Space

*3.3.1 Saliency Embedding.* Assume a pre-trained or fine-tuned CNN has a set of $L$ convolutional layers. For the image $i$ on the client $k$, the output of convolutional layer $l$ in GBP is denoted by $F_{i,l,k} \in \mathbb{R}^{C_l, H_l, W_l}$, where $C_l$ is the number of channels. The purpose of GBP is to compute the saliency latent variables of each image on each convolutional layer. Specifically, for the output $F_{i,l,k}$ of convolutional layer $l$, we calculate the feature maps as saliency latent variables $G_{i,l,k}$ in the following way:

$$G_{i,l,k} = \frac{\partial F_{i,l,k}}{\partial Y} \cdot \max(0, F_{i,l,k}), \tag{4}$$

where $Y$ is the model output (e.g., the target category score in a classification task). Then, we average the channel dimensions $C_l$ of saliency latent variables $G_{i,l,k}$:

$$\bar{G}_{i,l,k} = \frac{1}{C} \sum_{c=1}^{C} G_{i,l,k}^{(c)}. \tag{5}$$

*3.3.2 Adaptive Aggregation Adjustment.* For the saliency latent variable $\bar{G}_{i,l,k}$ of image $i$ at convolutional layer $l$, we denote the saliency of its features by the $L2$ paradigm:

$$N_{i,l,k} = \sqrt{\sum \|\bar{G}_{i,l,k}\|_2}. \tag{6}$$

We consider the feature maps from shallow to deep outputs to be valuable. Therefore, for each layer $l$, we assign a decaying weight $w_l$ that weights the feature map paradigms of the different layers:

$$w_l = \tau^{l-1}, \tag{7}$$

where $\tau$ is a hyper-parameter that defines the importance of going from shallow to deep, which we usually set to 0.5 as a rule of thumb. Then, the circling paradigms of all layers are summed to obtain saliency weight $R_k$ of client $k$, indicating the significance of the client $k$ in the aggregation:

$$R_k = \sum_{l=1}^{L} w_l \sum_{i=1}^{D} N_{i,l,k}. \tag{8}$$

Based on the above insight, we describe the procedure of calculating the aggregation weights $R_k$, called **saliency weight**, as shown in Algorithm 1. $P$ maps the input $x$ of image $i$ to the output $Y$, and $w_l$ is defined in Eq. 7. The whole process of Algorithm 1 is defined as **Guided Encoder**$(\cdot)$ in Algorithm 2, for each client $k$, whose output is the saliency weight $R_k$.

In the first round of aggregation, the model parameters are weighted and aggregated according to the saliency weight $R_k$ of client $k$. By displaying the amount of information about its saliency latent variables in Euclidean space, $R_k$ is able to provide a simple and intuitive generalization of the scale of image data features on a client. Thus, we understand saliency weight $R_k$ as a quantitatively representative coefficient of learnable features on client $k$. Then we can write the $r$-th round weighted aggregation as:

$$\Delta^{\mathrm{r}} = \sum_{k=1}^{K} \frac{R_k \Delta_k^{\mathrm{r}}}{\sum_{k=1}^{K} R_k}, \tag{9}$$

the $r$-th round global model can be calculated as follows:

$$\theta^{\mathrm{r}} = \theta^{\mathrm{r-1}} - \eta \Delta^{\mathrm{r}}. \tag{10}$$

---

**Algorithm 2** FedSLS: Exploring federated aggregation in saliency latent space

---

1: **Input:** training dataset $D_k$; initial parameter $\theta_0$; global communication rounds $R$; set of selected clients $S_k$; global and local learning rate $\eta_g, \eta_l$; attenuation coefficient $\tau$.
2: **Output:** global parameter $\theta_g$.
3: **Initialization:** Initialize $\Delta^0 = 0$ and $\theta^0$ as the global parameter at the server, then broadcasts to all clients.
4: **for** each client $k$ in parallel **do**
5:     *# pre-train or fine-tune for task adaptation*
6:     **for** each epoch **do**
7:         $\theta' \leftarrow$ Update $\theta^0$ on $D_k$.
8:     **end for**
9:     *# weights computation*
10:     $R_k =$ Guided Encoder$(\theta', \tau, D_k)$.
11: **end for**
12: *# training of federated*
13: **for** $r = 0, 1, \ldots, R-1$ **do**
14:     Sample subset $S_k \subseteq [K]$ of clients.
15:     **work on clients:**
16:     **for** each client $k \in S_k$ in parallel **do**
17:         client $k$ initialize the local parameter as $\theta_k^r$.
18:         $\theta_k^{r+1} \leftarrow$ Client Update$(\theta_k^r, x, \eta_l)$.
19:         $\Delta_k^r = \theta_k^{r+1} - \theta_k^r$.
20:         sends $\Delta_k^r$ and $R_k$ to server.
21:     **end for**
22:     **work on server:**
23:     *# aggregation with saliency weight*
24:     $\Delta^{\mathrm{r}} = \sum_{k=1}^{K} \frac{R_k \Delta_k^r}{\sum_{k=1}^{K} R_k}$.
25:     *# update global parameters and broadcast*
26:     $\theta^{\mathrm{r+1}} = \theta^{\mathrm{r}} - \eta^l \Delta^{\mathrm{r}}$.
27:     broadcast $\theta^{\mathrm{r+1}}$ to clients sampled in next round.
28: **end for**

---

### 3.4 Tractability of Optimization

One critical point in the procedures discussed above falls in the computation of saliency weights $R_k$ in Algorithm 1. Yet, updating $R_k$ dynamically before each aggregation round may impose considerable computational and memory demands. This process necessitates accessing saliency latent variables $G_{l,k}$ from each convolutional layer during BackPropagation, which, despite being executed once, requires substantial memory to store $G_{l,k} \in \mathbb{R}^{N,C,H,W}$.

For instance, using 1,000 images of CIFAR-10, the first layer's saliency latent variables $\sum_{i=1}^{1000} G_{i,1,k}$ demand approximately 11.76MB of memory, summing up to about 89.38MB for all layers. Extending this to CIFAR-10's full 50,000 image training dataset necessitates around 4,469MB across the federated system. Given the additional memory required for the model, optimizer, etc., the total memory usage escalates. This process will require 37 GFLOPs, so the computational effort associated with dynamic updates is not negligible. Updating the weights every round can impose a high computational and memory burden on the client and may bring stragglers [34] in

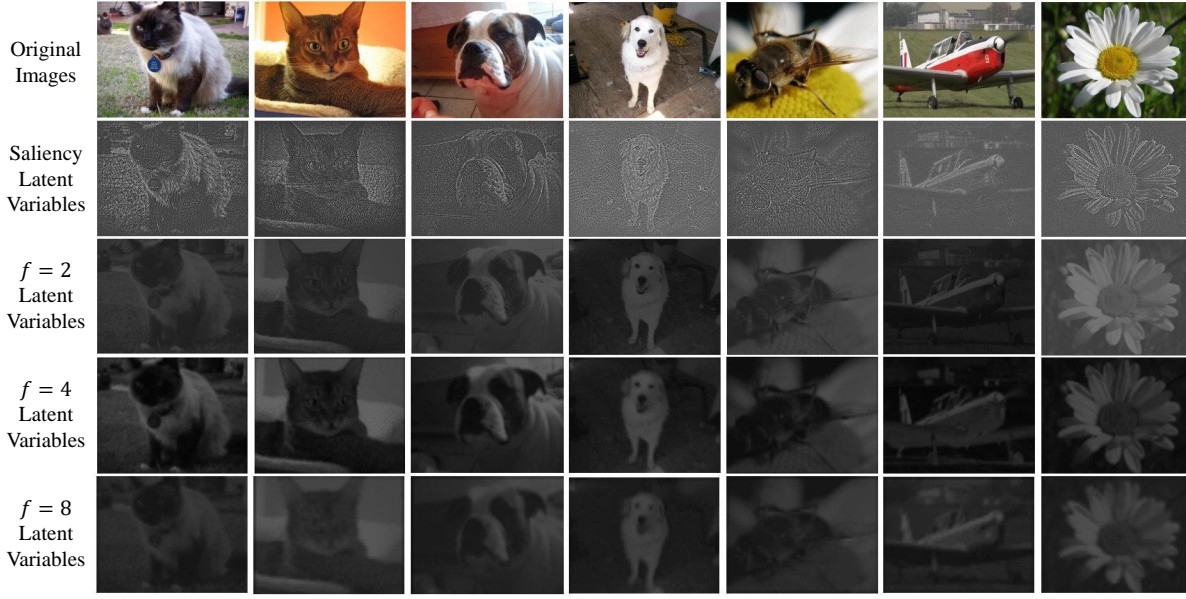

**Figure 3: The original images, saliency latent variables and three kinds of vanilla latent variables.**

federated learning system. In a real federated learning system, the data held by each client is often unbalanced. Therefore, during a communication round, the clients with less data have to wait for those with more data to update. This phenomenon will be exacerbated if the algorithm is updated dynamically, which can be very harmful to the performance of federated learning.

Fortunately, [2, 45] indicates that saliency feature maps do not differ significantly across classes, and the generative effect of the saliency feature maps is adjusted to the task. Therefore, we begin with a few preliminary training rounds on local data, fixing $R_k$ prior to the first aggregation and maintaining it constant thereafter:

$$R_k = \mathcal{S}\left(D_k, \theta_k^0\right). \tag{11}$$

In Section 4, our experiments comparing dynamic and static weighting strategies show comparable accuracy in most scenarios, with static weights performing slightly better in certain cases. This finding underscores the efficacy of our method, where saliency weight computation, streamlined to occur once, mitigates the overheads while aligning with task-specific adjustments.

To further reduce memory pressure, we average the channel dimensions of $G_{l,k}$ before saving the saliency latent variables. This significantly reduces the stored data:

$$\bar{G}_{l,k} = \frac{1}{C}\sum_{c=1}^{C} G_{l,k}^{(c)}, \tag{12}$$

where the saliency latent variables $G_{l,k}$ of the deeper level outputs of model may contain lots of channels, each representing a different feature. Averaging provides a comprehensive and informative saliency latent variable, reducing fluctuations due to randomness and local maxima. This will make the salient regions more stable and clear. See Algorithm 2 for details of FedSLS.

## 4 Experiments

### 4.1 Experimental Setup

*4.1.1 Datasets.* Following previous studies [22, 30], we compare the performance of FL algorithms on CIFAR-10/100 [20], CINIC-10 [4] and TinyImageNet datasets with 100 clients. The CIFAR-10 dataset consists of $50K$ training images and $10K$ testing images. All the images are with $32 \times 32$ resolution belonging to 10 categories. In the CIFAR-100 dataset, there are 100 categories of images with the same format as CIFAR-10. CINIC-10 extends CIFAR-10 with the addition of down-sampled ImageNet [5] images, consisting of $90K$ training images and $90K$ testing images. TinyImageNet includes 200 categories of $100K$ training images and $10K$ testing images, whose resolutions are $64 \times 64$. We use ResNet18 for CIFAR-10, CIFAR-100 and CINIC-10. For TinyImageNet, we adopt ResNet50 [10].

*4.1.2 Evaluation Measures.* We use the Top-1 Accuracy to evaluate the performance of methods:

$$Accuracy = (TP + TN)/(P + N),$$

where $P$, $N$, $TP$ and $TN$ are Positives, Negatives, True Positives and True Negatives, respectively.

*4.1.3 Baselines and Implementation.* We compare ours against several state-of-the-art federated methods focusing on aggregation scheme: **FedAvg** (AISTATS'17 [30]), **FedProx** (MLSys'20 [24]), **MOON** (CVPR'21 [22]), **FedDecorr** (ICLR'23 [36]), **FedAU** (ICLR'24 [42]). Referring to Section 3.2, we also designed the ablation experiment **FedLS**. In **FedLS**, the process is the same as in Algorithm 2 except for the feature embedding part. In reference to the method in [31], we replace the saliency latent variables of the **FedSLS** with the perceptually compressed latent variables obtained by 2×, 4× and 8× down-sampling. To reduce the memory cost, we also average these latent variables in the channel dimension. All other steps and processes are consistent with Algorithm 2. For all experiments,

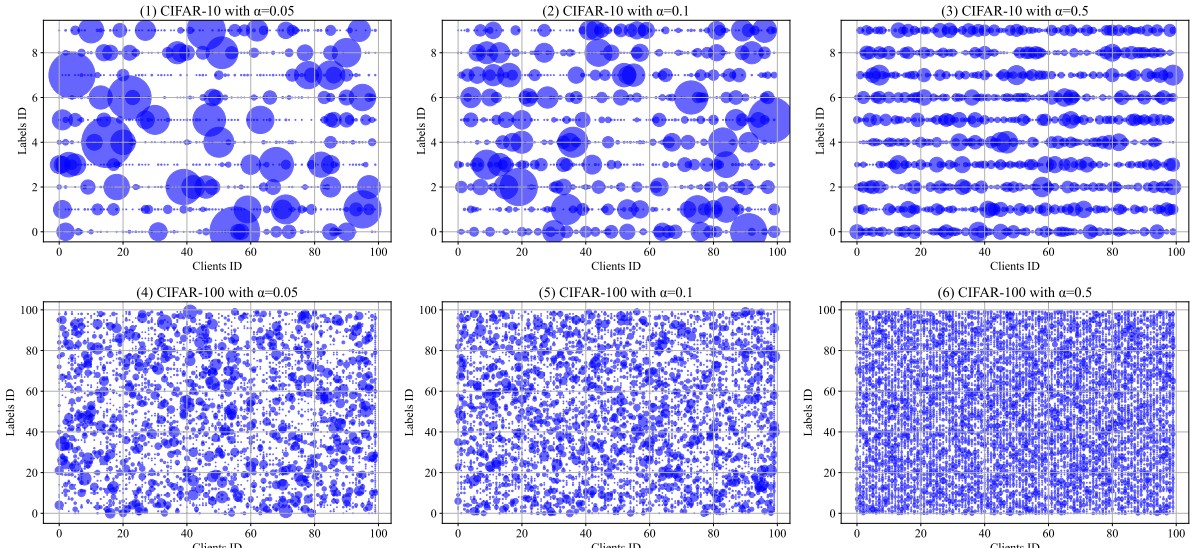

**Figure 4: Data heterogeneity among workers is visualized on CIFAR-10 and CIFAR-100, where the x-axis represents clients id, the y-axis represents the class labels on the training set, and the size of scattered points represents the number of training samples with available labels for that worker.**

including all baselines, FedSLS and FedSL, we load the pre-trained weights provided by PyTorch when initializing the model.

Following recent studies [22, 27, 32], all our experiments are performed on a centralized network with 100 workers. And we fix synchronization interval $I$ = 10, and the update will be a random selection of 10 clients out of a total of 100. For non-IID dataset partitioning over clients, we use Dirichlet-$\alpha$ (abbreviated as Dir($\alpha$)) sampling as [13], where the coefficient $\alpha$ measures the heterogeneity. In the experiments, we select the Dirichlet coefficient $\alpha$ from {0.05, 0.1, 0.5} for all datasets. A visualization of the data partitions for the CIFAR-10 at varying $\alpha$ values can be found in Figure 4. and the SGD optimizer with the learning rate $lr$ = 0.01. We set the communication round $T$ = 2,000 for CIFAR-10, CIFAR-100 and CINIC-10 datasets, $T$ = 1,000 for TinyImagenet datasets. The total number of epochs is 20,000 for CIFAR-10, CIFAR-100 and CINIC-10; 10,000 for TinyImagenet. The mean and standard deviation values for the average accuracy over the last 200 epochs were calculated from 10 experiments using 5 different random seeds and can be found in Table 1.

## 4.2 Performance Analysis

We display the results in Table 1. We observe that for all of the heterogeneous settings on all datasets, the highest accuracies are achieved by FedSLS. In particular, in the strongly heterogeneous settings where $\alpha \in \{0.05, 0.1\}$, FedSLS yields significant improvements of 2% to 20% over baselines on all datasets. The higher data heterogeneity, the more superior performance for FedSLS. On the other hand, for the less heterogeneous setting of $\alpha$ = 0.5, the impact of data heterogeneity is less significant, leading to smaller improvements from FedSLS. Such decrease in improvements is a

general trend and is also observed on other baselines. Our experiments in TinyImagenet were loaded with the ResNet50 pre-trained weights provided by PyTorch. Therefore, the difference between the experimental results in the three heterogeneous scenarios is not significant. Surprisingly, the accuracy of FedSLS still manages to be about 2% higher than baselines, which speaks volumes about the great contribution of FedSLS in dealing with data heterogeneity.

## 4.3 Ablation Study

This section focuses on the differences between saliency embedding and conventional embedding methods. We evaluated the visual fidelity of two types of embedding methods. Then, we designed experiments to replace the saliency latent variables with vanilla latent variables to guide the aggregation of federated learning.

*4.3.1 Experimental Setup.* Our method is closely related to the image, so we designed to compare vanilla latent variables and saliency latent variables with the original images. Specifically, we evaluate the ability of vanilla latent variables and saliency latent variables to extract features using three metrics: SSIM, Euclidean distance and cosine similarity. Considering generalization, we conduct experiments on a total of 10,000 images drawn from three publicly available classification datasets. We randomly draw 2,500 images of CIFAR-10 and CIFAR-100, respectively, and 5,000 images of Tiny-Imagenet. We performed a uniform pre-processing of the extracted images, normalized on the scale. We performed size, brightness and contrast normalization on the extracted images with a uniform size of $64 \times 64$. For the vanilla latent variables, we refer to the work of [31] for feature embedding using the down-sampling and define the parameters $f$ as 2, 4 and 8. For the saliency latent variables, we choose to use the ResNet18 for GBP. Unlike the performance

**Table 1: Performance comparison between FedSLS with baselines on CIFAR-10, CIFAR-100, CINIC-10, and TinyImagenet datasets. All algorithms were executed three trials with five different seeds, and the mean and standard derivation are reported.**

| Methods | | CIFAR-10 | | | CIFAR-100 | | | CINIC-10 | | | TinyImagenet | |
|---|---|---|---|---|---|---|---|---|---|---|---|---|
| | $\alpha = 0.5$ | $\alpha = 0.1$ | $\alpha = 0.05$ | $\alpha = 0.5$ | $\alpha = 0.1$ | $\alpha = 0.05$ | $\alpha = 0.5$ | $\alpha = 0.1$ | $\alpha = 0.05$ | $\alpha = 0.5$ | $\alpha = 0.1$ | $\alpha = 0.05$ |
| FedAvg (AISTATS'17) | 70.77±0.1 | 58.04±1.0 | 40.99±2.0 | 50.48±0.5 | 48.1±0.5 | 46.92±0.5 | 59.62±1.5 | 40.04±1.0 | 32.48±1.0 | 59.78±0.5 | 57.66±0.5 | 56.19±0.5 |
| FedProx (MLSys'20) | 73.64±2.7 | 61.22±0.3 | 41.64±1.5 | 50.17±1.3 | 48.73±0.4 | 46.41±0.3 | 59.81±0.9 | 40.64±2.7 | 33.97±1.2 | 60.20±0.1 | 59.66±0.4 | 57.16±0.3 |
| MOON (CVPR'21) | 77.02±0.8 | 62.79±1.7 | 55.64±2.4 | 50.63±1.4 | 48.62±0.2 | 45.48±0.3 | 62.91±1.3 | 41.86±1.1 | 40.54±1.3 | 61.31±0.90 | 60.75±0.6 | 59.21±0.4 |
| FedDecorr (ICLR'23) | 75.79±0.4 | 65.89±2.5 | 57.18±2.1 | 51.88±1.2 | 48.19±0.6 | 46.12±0.1 | 61.49±2.0 | 39.23±1.4 | 38.06±0.9 | 60.86±0.27 | 60.01±0.5 | 58.29±0.18 |
| FedAU (ICLR'24) | 74.91±0.6 | 60.04±1.8 | 49.24±3.8 | 51.49±1.5 | 46.5±1.5 | 47.08±1.5 | 59.96±1.0 | 38.96±1.0 | 33.04±1.0 | 61.09±0.4 | 59.81±0.4 | 57.43±0.6 |
| FedSLS | **82.67±0.2** | **70.51±3.6** | **63.43±3.6** | **52.44±1.5** | **49.6±0.5** | **47.79±1.5** | **65.43±2.5** | **44.79±2.6** | **43.22±2.6** | **62.73±0.5** | **61.47±0.5** | **59.49±0.5** |
| FedSL $f = 2$ | 79.86±0.3 | 64.97±0.5 | 38.59±0.7 | 51.64±0.3 | 49.4±0.3 | 46.25±0.3 | 69.48±0.2 | 40.15±1.4 | 30.87±2.5 | 61.68±0.2 | 59.33±0.3 | 57.34±0.8 |
| FedSL $f = 4$ | 81.65±0.3 | 48.91±0.8 | 38.44±0.4 | 52.19±0.5 | 48.52±0.5 | 42.98±0.5 | 69.54±0.8 | 42.27±1.4 | 31.35±1.9 | 61.65±0.2 | 59.28±0.6 | 58.00±0.3 |
| FedSL $f = 8$ | 82.31±0.2 | 56.98±0.4 | 50.65±0.4 | 51.28±0.2 | 48.94±0.3 | 45.13±0.4 | 64.96±0.2 | 42.52±1.6 | 37.84±1.1 | 61.37±0.2 | 59.06±0.3 | 56.85±0.4 |

comparison experiment mentioned above, we only take the output image of the first layer of BackPropagation in this experiment. We save both vanilla latent variables and saliency latent variables locally and read them when needed. When generating these images, we average over the channel dimensions as before, so we save all grey-scale maps. In order that the later evaluation metrics are not affected, we similarly normalize all images in terms of size, brightness and contrast. The Figure 5 shows all results, and Figure 3 shows different kinds of latent variables.

*4.3.2 SSIM.* Structural similarity index (SSIM) [43] is a metric used to measure the similarity between two images, taking into account luminance, contrast and structural information, and provides an excellent reflection of human perception of image quality. If the images obtained through saliency embedding are most similar to the original images, it suggests that we are proficient at perceiving and extracting features to a certain extent.

The SSIM results of the four methods with the original images are presented on the far left of Figure 5. It is evident that the median and mean of the saliency latent variables are the highest among the four. This suggests that the saliency latent variables can retain more information from the original images in the image representation. The median latent variables for 8× down-sampling are the lowest, while those for 2× down-sampling are significantly higher than the other two. This also indicates that as the degree of image compression increases, the more feature information is lost from the image and hence the similarity with the original image decreases. It is also observed that the data distribution to which the saliency latent variables belongs is narrower, indicating that the quality of the images obtained by the GBP method is relatively consistent and less fluctuating. While the other box plots show wider interquartile spacing, especially 8× down-sampling, indicating that the image quality fluctuates more. And the outliers (small black circles) mainly appear at lower SSIM scores, especially more pronounced at higher compression ratios $f$.

*4.3.3 Euclidean Distance.* The Euclidean distance is one of the most straightforward measures of distance. We find the Euclidean distance between the original image and the latent variable to

represent its similarity. The two methods are inherently similar and we use this to corroborate each other and eliminate chance.

The results of their comparison we show on the far right of Figure 5, where we can see that the situation is similar to the results of SSIM. The median and the mean of the significance plot are the smallest, indicating that the Euclidean distance between the significance plot and the original image is the smallest. There are a large number of outliers on the box plots of all the methods, but the distribution of outliers in the significance plot is much narrower, which indicates that the variability of the significance plot with respect to the original image is relatively smaller among the four methods, and the stability of this method is better. Compared to the 2× and 4× down-sampling, the 8× down-sampling has fewer outliers, but the median and mean are relatively higher. This could mean that as the encoder strength increases (larger down-sampling factor), the overall variability of the images increases, although it can provide more consistent results in some cases. It can be inferred that the saliency latent variables extract more subtle features with less variability.

*4.3.4 Cosine Similarity.* The vanilla latent variables obtained by down-sampling are themselves compressed and smaller compared to the original image, and the two previous methods of comparison require the alignment of features such as scale, brightness, contrast and color to reduce the interference of external factors. For example, the latent variables obtained by 8× down-sampling may be enlarged in size by linear interpolation, and some information will be lost in the process. Therefore we still need more objective methods to prove our point.

Fortunately, the work of [33] points out that cosine similarity can be utilized for feature evaluation. Cosine similarity, as highlighted by [33], serves as a robust metric for feature evaluation in image processing and deep learning. It computes angular distances between vectors, disregarding their magnitudes, thereby facilitating the comparison of feature orientations in high-dimensional spaces without necessitating pre-processing adjustments. Cosine similarity's efficacy stems from its ability to discern feature directionality, where a proximity to 1 denotes high similarity.

So we design to input the original image and four latent variables into ResNet50, and then use the output of the model to calculate

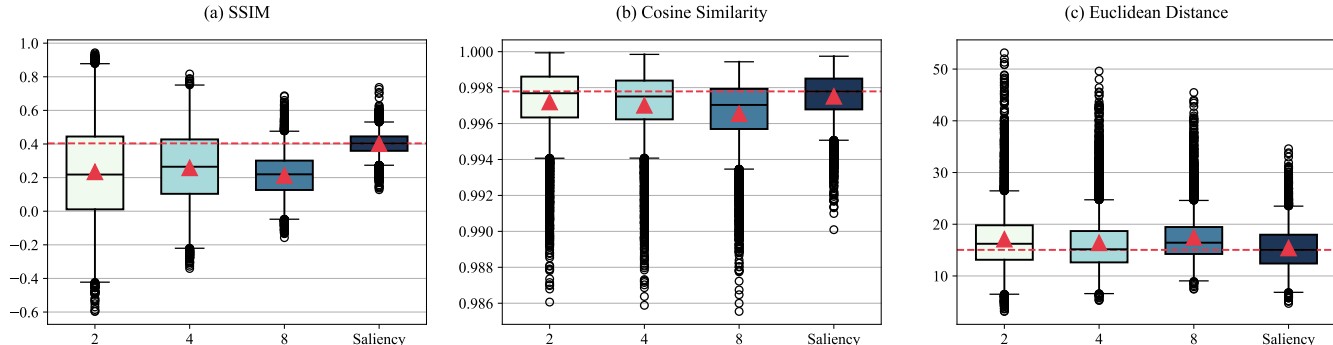

**Figure 5: Results of SSIM, Cosine similarity and Euclidean distance scores for vanilla latent variables and saliency latent variables, where the vanilla latent variables are derived from 2×, 4× and 8× down-sampling. (a) Saliency latent variables have the better SSIM scores, the higher medians and means, and narrower boxes and whiskers. (b) Performance of cosine similarity scores clearly shows that the saliency latent variables have less variance and more similarity. (c) The saliency latent variables has the smallest Euclidean distance from the original image, indicating a greater similarity to the original image.**

the cosine similarity respectively. We have displayed the results in the middle of Figure 5, where we can see that the median and mean of the saliency latent variable are higher than the other three by a narrow margin, but the box corresponding to the saliency latent variable is narrower and the whiskers are shorter, which proves that the variance of the cosine similarity scores is smaller, the scores are more consistent, and the indicated features are more reliable than the others'. In addition, the outliers of the saliency latent variable are also more concentrated than those of the other three, and the saliency latent variable representation is relatively more stable. So we can conclude that the cosine similarity scores of the saliency latent variables are integrally better. This is enough to show that GBP effectively emphasizes the features that the model considers most informative.

*4.3.5 Comparisons in Federated Learning.* To further prove our point, we designed to replace the saliency latent variables with vanilla latent variables in the framework of FedSLS. As with the other baselines in Table 1, we conduct experiments on each of the four datasets using vanilla latent variables obtained from the three down-sampling multiplicities. Refer to Section 4.1 for additional details on the setup of the experiments.

Again, the results are shown in Table 1. The results remain that FedSLS is at state-of-the-art in each experimental setup. In the less heterogeneous settings where $\alpha = 0.5$, FedSL results are only one line worse than FedSLS, above many baselines. However, in the strongly heterogeneity setting $\alpha \in \{0.05, 0.1\}$, the accuracy of FedSL drops rapidly and performs very poorly. We believe this is due to the fact that these vanilla latent variables do not extract the features needed for learning very efficiently and also introduce redundancy and irrelevant information interference. This is because conventional feature embedding methods pay more attention to overall features than local details during feature extraction, and these features may be links between the subject instance and the background and environment. These features with little learning value influence the server's weight assignment to individual clients. Therefore, in strongly heterogeneous environments, referring to

the results in Figure 5 , the effect of some outliers on the clients may be amplified as the weight value increases. However, FedSL can still outperform most baselines in the less heterogeneous environment $\alpha = 0.5$, which shows that our strategy of mining image saliency features in latent space to adaptively adjusting aggregation is effective.

*4.3.6 Summary.* Our findings indicates that saliency latent variables retained more original image information, evident from higher median and mean SSIM values and narrower interquartile ranges, suggesting consistent quality. Euclidean distance mirrored these results, with saliency latent variables showcasing minimal variability and greater stability. Cosine similarity confirmed the superiority of saliency latent variables, indicating that GBP effectively accentuates model-deemed informative features. Lastly, within the FedSLS framework, replacing saliency latent variables with vanilla latent variables substantiated the inefficiency of conventional feature embedding in heterogeneous environments, further demonstrating the validity of the significance latent variable.

## 4.4 CONCLUSION

In this paper, we propose a saliency latent space feature aggregation method (FedSLS) across federated clients. By Guided BackPropagation (GBP), we transform deep models into powerful and flexible visual fidelity encoders, applicable to general state inputs across different image domains, and achieve powerful aggregation in the form of saliency latent features. To that effect, we experimentally demonstrate the superiority of the Saliency Embedding over conventional feature embedding methods. The extensive experimental validation of FedSLS across various datasets establishes its superiority, showcasing remarkable performance enhancements and state-of-the-art results, particularly in environments characterized by severe data heterogeneity. The findings underscore the potential of FedSLS to substantially contribute to the optimization of aggregation weights in federated learning, offering a robust solution to the complexities introduced by non-IID and imbalanced data heterogeneity distributions.

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
