# OpenReview forum: "FedSLS: Exploring Federated Aggregation in Saliency Latent Space"
_acmmm.org/ACMMM/2024/Conference — MM2024 Poster_

### Official Review · Reviewer_KdWM · 2024-05-09

**Rating:** 4
**Confidence:** 3

**Summary:**

This paper propose a saliency latent space feature aggregation method across federated clients for coping with slow convergence be resulting from data heterogeneity. Deep models are transformed into powerful and flexible visual fidelity encoders to  achieve powerful aggregation in the form of saliency latent features.

**Strengths:**

FedSLS not only outperforms existing state-of-the-art federated learning methods.

**Limitations:**

the word size in Figure 2 are too bold;

the following text appears in both abstract and conclusion.
In this paper, we propose a saliency latent space feature aggregation method (FedSLS) across federated clients. By Guided ackPropagation
 (GBP), we transform deep models into powerful and ffexible visual ffdelity encoders, applicable to general state inputs across different image domains, and achieve powerful aggregation in the form of saliency latent features.

**Suitability:**

2

---

### Official Review · Reviewer_fHzS · 2024-05-23

**Rating:** 2
**Confidence:** 3

**Summary:**

This paper suggests a saliency latent space feature aggregation method (FedSLS) in the federated setting. It states that this paper is the first attemtp to transform deep models into high visual fidelity encoders with GBP.

**Strengths:**

(1) The use of GBP can balance the computational complexity and the details.
(2) Even in environments characterized by severe data heterogeneity, the  superiority of FedSLS is also validated.
(3) This paper investigates cross-client consistency representations in the latent space.

**Limitations:**

(1) The core idea is to map the fatures into a latent space. The innovation of this idea is limited.
(2) The GBP is not the original contribution of this paper, thus, the contribution of this paper is limited.
(3) The experiments should be extended.
(4) The number of comparative approaches in the experiments should be increased.
(5) Feature visualization after mapping is necessary.

**Suitability:**

2

---

### Official Review · Reviewer_qTy9 · 2024-05-25

**Rating:** 4
**Confidence:** 2

**Summary:**

Federated Learning contends with client data heterogeneity. Prior methods applied various aggregation strategies, which either increased memory demands or resulted in suboptimal generalization on certain clients. This paper introduces a method to transform models into high visual fidelity encoders that strike an optimal balance between minimizing complexity and maintaining detail. The authors also introduce a novel aggregation approach to assess cross-client consistency. The paper is well-structured, with clear contributions and compelling experimental results.

**Strengths:**

- The clarity of the paper makes it accessible and comprehensible.
- The experimental results demonstrate notable improvements over recent methods.

**Limitations:**

- The caption in Figure 4 should provide a more detailed explanation of data heterogeneity, as the current figures illustrating data distribution are somewhat unclear and challenging to interpret.
- The authors assert that traditional feature encoders produce a semantic representation but fail to filter out irrelevant features. More rigorous experiments are needed to substantiate this claim. While Figure 3 might illustrate this issue to some degree, the paper does not sufficiently explain how this figure demonstrates the presence of “irrelevant features” in conventional encoders. Further clarification is necessary on how the proposed method addresses this issue and why a more distinct outline of Saliency Latent Variables leads to enhanced final results.
- The authors claim that previous aggregation strategies lead to poor generalization. They should clearly explain how their experiments demonstrate the improved generalization of their proposed method.

**Suitability:**

2

---

### Official Review · Reviewer_PPRp · 2024-05-27

**Rating:** 4
**Confidence:** 3

**Summary:**

FedSLS utilizes Guided BackPropagation (GBP) technology, combining the model's output with the gradient of the feature map to accurately capture the salient features most important for model prediction. Based on the importance of these salient features, it adaptively adjusts the aggregation weights to ensure that the critical features from each client have sufficient influence on the global model.
Experimental results show that FedSLS significantly outperforms existing state-of-the-art methods in handling highly heterogeneous data environments.

**Strengths:**

FedSLS utilizes GBP technology to combine the model's output with the gradient of the feature map, not only retaining positive features but also highlighting the most important activation features for the model, thereby extracting salient features. Additionally, the calculation of saliency weights considers the multi-level structure of features, assigning greater weight to shallow saliency features, further enhancing the contribution of important information. Finally, it adaptively adjusts the aggregation weights based on the importance of the saliency features, ensuring that the critical features from each client are used to update the global model.
By aggregating salient features, FedSLS is better able to capture the key information in the data from different clients, enhancing the model's generalization ability, and ensuring that it maintains good performance across various data distributions.

**Limitations:**

1. The calculation of saliency weights requires computing the gradient of each feature map and considering multi-level structural features. When handling large datasets and deep neural networks, this overhead may significantly increase.
2. The proposed method in this paper and the comparison methods do not use a unified network backbone during training, failing to eliminate the impact of the inherent feature extraction capabilities of neural networks on the experimental results.
3. Although this method shows significant improvements in highly heterogeneous settings, the performance gains are less pronounced in less heterogeneous environments (α = 0.5). This indicates that the method is more effective under certain conditions, which may limit its general applicability.
4. FedSLS primarily focuses on data heterogeneity and has not explored the randomness and complexity of client participation patterns in the experiments.

**Suitability:**

2

---

### Meta-Review · Area_Chair_ckCg · 2024-06-30

**Recommendation:** Accept (Poster)
**Confidence:** 4

**Metareview:**

The paper introduces a novel saliency-guided algorithm for federated learning, receiving positive feedback from three reviewers. Despite some concerns raised by reviewers regarding the originality of the idea and the depth of empirical evaluations, the authors' rebuttal, while not fully convincing to all reviewers, provided insights that supported the merits of their work. ACs considered the confidential comments sent by the authors in the post-rebuttal discussions and their final decision.

Considering the overall judgment of the reviewers and acknowledging the potential contributions of the proposed algorithm, the ACs recommend accepting the paper. Congratulations to the authors on this achievement. It is advised that the authors carefully consider the feedback provided by the reviewers, particularly regarding the originality and depth of empirical evaluations, and ensure that promised enhancements and clarifications from the rebuttal are incorporated into the final submission.